# New Insights and Potential Therapeutic Targeting of CB2 Cannabinoid Receptors in CNS Disorders

**DOI:** 10.3390/ijms23020975

**Published:** 2022-01-17

**Authors:** Berhanu Geresu Kibret, Hiroki Ishiguro, Yasue Horiuchi, Emmanuel S. Onaivi

**Affiliations:** 1Department of Biology, College of Science and Health, William Paterson University, Wayne, NJ 07470, USA; 2Department of Neuropsychiatry and Clinical Ethics, Graduate School of Medical Science, University of Yamanashi, Chuo, Yamanashi 409-3898, Japan; hishiguro@yamanashi.ac.jp; 3Department of Psychiatry and Behavioral Sciences, Tokyo Metropolitan Institute of Medical Science, Tokyo 156-8506, Japan; horiuchi-ys@igakuken.or.jp

**Keywords:** endocannabinoidome, endocannabinoid system, CB2R, neuropsychiatry, Alzheimer’s disease, schizophrenia, anxiety, Huntington’s disease, addiction

## Abstract

The endocannabinoid system (ECS) is ubiquitous in most human tissues, and involved in the regulation of mental health. Consequently, its dysregulation is associated with neuropsychiatric and neurodegenerative disorders. Together, the ECS and the expanded endocannabinoidome (eCBome) are composed of genes coding for CB1 and CB2 cannabinoid receptors (CB1R, CB2R), endocannabinoids (eCBs), and the metabolic enzyme machinery for their synthesis and catabolism. The activation of CB1R is associated with adverse effects on the central nervous system (CNS), which has limited the therapeutic use of drugs that bind this receptor. The discovery of the functional neuronal CB2R raised new possibilities for the potential and safe targeting of the ECS for the treatment of CNS disorders. Previous studies were not able to detect CB2R mRNA transcripts in brain tissue and suggested that CB2Rs were absent in the brain and were considered peripheral receptors. Studies done on the role of CB2Rs as a potential therapeutic target for treating different disorders revealed the important putative role of CB2Rs in certain CNS disorders, which requires further clinical validation. This review addresses recent advances on the role of CB2Rs in neuropsychiatric and neurodegenerative disorders, including, but not limited to, anxiety, depression, schizophrenia, Parkinson’s disease (PD), Alzheimer’s disease (AD), Huntington’s disease (HD) and addiction.

## 1. Introduction

With the significant progress and advances in cannabis and cannabinoid research, there is a renewed focus and interest in targeting components of the ECS in CNS disorders, even in children, and in neurodegenerative disorders. Cannabinoids are naturally occurring compounds found in the *Cannabis* plant species. Delta-9-tetrahydrocannabinol (Δ^9^-THC) is the most abundant psychoactive, and most studied cannabinoids among the over 500 different compounds present in the plant that produces their effect through binding cannabinoid and non-cannabinoid receptors [1]. The endocannabinoid system (ECS) is composed of the endogenous ligands, endocannabinoids (eCBs) anandamide (AEA) and 2-arachidonoylglycerol (2-AG), their synthesizing and metabolizing enzymes, and the cannabinoid receptor type 1 (CB1R), type 2 (CB2R), and other putative cannabinoid receptor candidates. The ECS is an endogenous signaling system involved in regulation and homeostasis in some physiological processes in the body, including the neuro-immune communication between cells, appetite and metabolism, memory, and more. The expanded ECS, which includes different protein and lipid mediators, is involved in the pathophysiology of mental disorders, and targeting this system might be of benefit in the management of a number of these diseases [2].

There are two canonical cannabinoid receptors (CBRs), CB1R and CB2R, that are G-protein-coupled receptors (GPCRs) [3]. CB1Rs are the most abundant GPCRs in the brain, and are expressed in the basal ganglia nuclei hippocampus, cerebellum, neocortex [4], and other brain regions. Higher levels of CB2Rs are found predominantly in peripheral organs with immune function [5,6,7]. The previous notion was that CB2Rs are absent in the brain [7,8,9]; however, a large body of evidence now demonstrates CB2R expression in microglia and neurons in the hippocampus, striatum, and brain stem [10,11]. They are also expressed in mouse brain dopamine neurons and are involved in different physiologic and pathologic conditions [12].

There is a body of literature on the effects and the role of CB1Rs in the central nervous system (CNS), whereas the contribution of CB2Rs in neuropsychiatry is not well documented, since these receptors were previously considered to be found only in the “periphery” and were thought to be absent in the brain [13]. However, CB2R expression in cells of immune origin and their up-regulation has been associated with CNS disorders that are linked with underlying neuroinflammation. In addition, the recognition and knowledge regarding cannabinoids which lack CNS intoxicating effects is adding to the growing awareness of the putative role of CB2Rs in different disorders [2].

Therefore, in this narrative review method, we have summarized our pivotal research contribution to the discovery of the CB2Rs, with a focus on the preclinical and clinical work of other researchers using cannabis, cannabinoids, CB1R, CB2R, and cannabinoid receptor genetics in neurological and psychiatric disorders for PubMed searches. The articles not written in English and those outside of this subject were excluded. It is noteworthy that the expression of CB2Rs in the brain attracted significant attention, debate, and controversy [14]. The controversy was that CB2Rs are restricted to peripheral tissues and are found predominantly in cells of immune origin. Unlike CB1Rs, CB2Rs were thus initially considered “peripheral” cannabinoid receptors (CBRs), because Munro et al. 1993 and later many investigators were not able to detect neuronal CB2R expression in the brain. Therefore, there has been continuous debate and controversy about the functional neuronal CB2Rs. Some reported that CB2Rs were absent from the brain or that CB2R is a cannabinoid receptor with an identity crisis, and others described CB2R as a sphinx wrapped in a mystery. Following our discovery of the presence and functional expression of CB2Rs in the brain [15,16,17,18], other studies have overwhelming confirmed that functional CB2Rs are present in neurons. Nevertheless, this debate and controversy about functional neuronal CB2Rs lingers, despite the overwhelming body of clinical and basic research, using a number of molecular, electrophysiological, ultrastructural, imaging, and determination of the crystal structures of both CB1R and CB2R, indicating the functional neuronal expression of CB2Rs. Thus, this review summarizes the therapeutic role of CB2Rs in neuropsychiatric and neurodegenerative disorders such Alzheimer’s disease (AD), anxiety, depression, schizophrenia, Parkinson’s disease (PD), addiction, Huntington’s disease (HD), and other CNS disturbances.

## 2. Overview of the ECS and New Insights into the Endocannabinoidome-eCBome

The ECS is composed of the two canonical receptor subtypes CB1R and CB2R, eCBs, and their synthesizing and degrading enzymes [4,19]. eCBs are endogenous substances that produce a physiological effect through binding to CBRs. The most characterized and studied eCBs are AEA and 2-AG [20]. CB1R and CB2R are well characterized and belong to the large family of the GPCRs. They differ in their signaling mechanisms and tissue distribution. Both CB1Rs and CB2Rs block adenylyl cyclase (AC) and stimulate p42/p44 mitogen-activated protein kinase (MAPK) activity [3,21]. The activation of CB1Rs but not CB2Rs results in the release of inositol triphosphate and intracellular Ca^2+^. CB1R activation also results in a regulation of the inwardly rectifying potassium currents (Kir) through the βγ-subunits [21]. There are also reports showing the coupling of CB1Rs to other G proteins, including Gs, although with low efficacy, resulting in the stimulation of AC [22,23]. CB1Rs are the most abundant GPCRs in the brain, expressed in the basal ganglia nuclei, hippocampus, cerebellum, and neocortex. CB1Rs are localized in pre-synaptic terminals at very high levels on GABAergic and at reduced levels on glutamatergic neurons [4,24]. They are also highly expressed in dopamine type-1 receptors (D1Rs) expressing neurons in the brain [25]. CB1Rs are responsible for modulating different emotional behavior and cognitive activities, and the alteration of these receptors might be involved in different neuropsychiatric disorders [26,27,28]. CB2Rs were previously thought to be located only in the periphery [5,6,7] and were believed to be absent in the brain [7,8,9]. Contrary to these previous notions, a number of studies demonstrates CB2Rs expression on microglia and neurons in the mid brain region and the brain stem [10,11,29]. CB2Rs which are found in dopamine neurons play a significant role in psychosis, drug reward, eating disorders, depression, addiction, autism spectrum disorders, and synaptic plasticity [12]. As our understanding of the involvement of the ECS in different CNS physiologic and pathologic conditions increases, the ECS has gained a special attention as a potential therapeutic target and in designing safe and effective drugs to treat CNS disorders [30]. Furthermore, understanding the underlining mechanisms for the involvement of the ECS in different pathological conditions, including neuropsychiatric disorders, is important in the early diagnosis and treatment of these disorders.

New insights to an expanded ECS: the endocannabinoidome (eCBome) has broadened, targeting the complex eCBome in CNS disorders at a higher level than previously estimated. This system includes N-acylethanolamines (NAEs), 2-acylglycerols (2-AcGs), N-acyl-aminoacids, N-acyl-dopamines, and N-acyl-serotonins [2]. As studies unravel the complexity of this system, the interest in utilizing the eCBome system as a potential therapeutic target for the treatment of neuropsychiatric disorders is increasing [1,2]. In addition, previous studies revealed that targeting the CB1R is associated with adverse effects including anxiety, depression, and even suicidal ideation [31,32], and hence the search for new therapeutic targets with minimal adverse effects is gaining special attention. Recent studies showed that targeting the CB2Rs, unlike the CB1Rs, is safe and effective and that CB2Rs might be new possibilities for a safe targeting of the eCBome system [33]. An emerging microbiome and eCBome crosstalk is revealing a bidirectional gut–brain communication. Our discovery of functional neuronal CB2Rs (15-19) and reports of enhanced CB2Rs during inflammation have raised questions regarding their role in regulating neuroinflammation and behavior. Neuroinflammation encompasses a wide array of cellular processes, including the activation of microglia and astrocytes, enhanced pro-inflammatory cytokines, chemokines, eicosanoids, complement activation, and acute phase proteins. These processes have also been proposed to underlie the pathophysiology of several neuropsychiatric disorders, including AD, PD, HD, and other CNS disturbances. Neuroinflammation is emerging as a key component in the effects of CB2Rs expressed in macrophages, microglia, astrocytes, and neurons that are important regulators of the immune response. Therefore, more studies will undoubtedly identify the role of CB2Rs as possible targets in CNS disorders, associated with neuroinflammation for neuroprotection in modulating the bidirectional eCBome–gut microbiome axis.

## 3. Unexpected CNS CB2R Expression and Function

CB2R is located on human chromosome 1p36 [5,34,35,36] and mouse chromosome 4. Earlier studies, when it was first cloned in 1993, could not detect CB2R in the brain tissue and suggested that CB2Rs were absent in the brain, and they were considered as “peripheral” CBRs [5,6,37,38]. However, recent studies identified the presence of CB2Rs throughout the CNS using different methods, including in situ hybridization (ISH), immunostaining, and gene expression [19,39,40,41,42]. Unlike the well characterized functional expression of CNS CB1Rs, the neuronal expression of CB2Rs in the CNS has been much less well established and characterized [43]. However, some previous studies reported the discovery and functional characterization of CB2Rs in neural progenitor cells, neurons, and glial and endothelial cells [10,17,18,44,45]. Compared to CB1Rs, CB2Rs are less expressed in the brain, and hence their physiological role in the brain has not been as well investigated as that of CB1Rs. CB2Rs are less expressed in the brain, suggesting that they are not involved in the adverse effects that are observed after the activation of CB1Rs [46]. The activation of CB2Rs inhibits AC activity and initiates MAPK and phosphoinositide 3-kinase (PI3K)-Akt pathways (Figure 1), with a subsequent activation of the Jun N-terminal protein kinase (JNK), extracellular signal-regulated kinase (ERK)1/2, and p38 [47,48]. In addition, CB2R agonists result in an increased synthesis of ceramide, a sphingolipid messenger, particularly in tumor cell lines, which induces apoptotic cell death [49]. CB2Rs are mainly expressed post-synaptically, and their activation inhibits the postsynaptic neuronal function through membrane potential hyperpolarization [50].

While a large body of evidence documents how the *CNR1* gene is regulated, there is a limited characterization of the *CNR2* gene, with a limited characterization of the structure, regulation, and variations of the *CNR2* gene. Likewise, the physiological and pathological role of the CB2Rs in the brain as well as in the periphery has not been well investigated and characterized when compared to CB1Rs. However, due to recent advances and our increasing understanding of the signaling mechanisms of the CB2Rs, the functional role of these receptors is gradually emerging [43]. CB2Rs are involved in modulating a variety of behavioral effects in the CNS. It has been reported that CB2Rs modulate food intake, body weight [51,52], depression and anxiety [12,53], drug addiction [54,55], and schizophrenia-like behavior [56]. Brain CB2Rs are expressed at low levels under physiological conditions; however, in pathological conditions, such as neuropathic pain [57], stroke [58], traumatic brain injury [59], neurodegenerative diseases [54,60,61], or drug addiction [62,63], their expression is enhanced and up-regulated. The inducible nature of CB2Rs during events with underlying inflammation can make them a potential therapeutic target, and ligands that activate or inhibit the activity of CB2Rs might be used to treat different disorders without causing profound adverse drug and intoxicating effects [46].

## 4. Role of CB2Rs in CNS Disorders

Advances in preclinical research in animal models using different molecular techniques—like Western Blotting, immunohistochemistry, gene expression and a battery of behavioral tests—have identified the involvement of elements of the ECS, particularly CB2Rs (Figure 2), in models of CNS function and dysfunction. Previous studies reported evidence for the involvement of CB2Rs in neuropsychiatry [16,17,18,42,45,64,65]. As the clinical and functional implications of neuronal CB2Rs in the brain is gradually becoming clearer, more research is unraveling their contribution in neuropsychiatry and global drug addiction [38] in this era of COVID, due not only to the lockdown and isolation, but also to the increased drinking behavior and opioid overdose and deaths, particularly with the increased opioid epidemic [66]. Currently there is no effective therapy or cure for most neuropsychiatric disorders. Besides, the available conventional therapies are associated with a wide range of unwanted adverse drug effects. Hence, there is a need for new therapeutic targets for the development of safe and effective medications to prevent or retard the disease process in neuropsychiatric disorders [67]. The studies reviewed in the previous sections demonstrated the role of CB2Rs in the regulation of physiologic functions, and hence they can be utilized as a potential target in neuroinflammatory and neurodegenerative disorders [68,69], but require further clinical trials. Below, we briefly review the evidence supporting the putative role of CB2Rs in various neuropsychiatric and neurodegenerative conditions. The role of CB2Rs in these disorders are summarized in Table 1 and Table 2.

### 4.1. Role of CB2Rs in Psychiatric Disorders

#### 4.1.1. Anxiety-Related Disorders

Anxiety is a CNS disorder afflicting all age groups and associated with possible threats that provides a coping mechanism to reduce contacts and avoiding stressful encounters [70]. The ECS plays a pivotal role in the regulation of mood disorders [71] and CB2Rs are expressed in the amygdala, hippocampus, prefrontal cortex, and hypothalamus. These brain areas are involved in regulating anxiety-like behavior, and this suggests the potential role of CB2Rs in the regulation of emotional responses [10,18,40,72]. We reported polymorphisms in both CB1R and CB2R in neuropsychiatric disturbances and using animal models of CNS disorders. Using the a two-compartment black and white chamber and an elevated plus maze test for assessing anxiety-like behavioral responses, we have reported on the effect of CB2R ligands on stress-induced anxiety-related behavior in mice. Our result showed that the acute administration of JWH-015 (1–20 mg/kg), which has more affinity for binding to CB2Rs than to CB1Rs, dose-dependently induced an anxiogenic response in the black and white box but attenuated the stress-induced gender-specific aversion to the open arms of the elevated plus maze [18,42]. Using rodent models of acute and chronic anxiety-like behavioral responses, Valenzano et al. [73] demonstrated that the administration of GW405833 (100 mg/kg), a CB2R agonist, induced anxiolytic-like effects in the marble-burying test that were not reversed by the administration of a CB2R antagonist. However, in a mouse stress model, chronic mild stress (CMS) increased anxiety-like behavioral responses in the Zero maze test paradigm, and the treatment with JWH-015 (20 mg/kg) reduced the anxiety-like behavioral effects similar to the effects of the antidepressant fluvoxamine [74].

The involvement of CB2Rs in anxiety-like disorders was described by studies using selective CB2R antagonists. The acute administration of AM630 (1, 2, or 3 mg/kg) in mice induced anxiogenic-like behavioral effects, whereas with chronic administration anxiolytic-like effects were reported using the light-dark box and the elevated plus maze test [53]. Treatment with AM630 (3 mg/kg) worsened the anxiety-like behavioral responses induced by CMS, contrary to the effects observed with JWH-015 treatment, as described above [74]. Interestingly, studies using transgenic mice overexpressing CB2R in the CNS (CB2xP mice) showed no effects against anxiogenic-like stimuli in the light-dark box and the elevated plus-maze behavioral tests [53]. In agreement with this, another study using mice lacking the CB2R gene (*Cnr2*−/− mice) revealed that the CB2R gene knockout mice developed higher levels of anxiety-like behavioral responses in both tests [56]. We have investigated the involvement of CB2Rs in an emotionality test using dopamine neuron cell-type specific CB2R conditional knockout mice (DAT-*Cnr2* cKO). The study utilized a battery of behavioral tests [12], and the results demonstrated that the DAT-*Cnr2* cKO mice experience significant anxiety-like behavior compared to the wild-type mice. Clinical studies on the role of CB2Rs on anxiety-like behavior are very scarce. However, one study conducted on children with a primary anxiety disorder investigated whether genetic variations in the ECS explained individual differences in response to cognitive behavioral therapy [75]. The study revealed that there is a relationship between the rs2070956 polymorphism of the *CNR2* gene and a reduced treatment outcome in children with anxiety disorders. While the rs2070956 and rs2500413 polymorphisms are functionally unknown, rs2500413 is located in the main exon of CNR2 gene.

The results obtained from animal and clinical studies show a role of CB2Rs in modulating anxiety-like behavior, but further studies are needed to determine the therapeutic potential of CB2Rs in anxiety disorders.

#### 4.1.2. Depression and CB2R-ECS

As neuroinflammation is increasingly linked to a number of CNS disorders, and microglia cells, the brain’s intrinsic immune cells, are the primary contributors to neuroinflammatory responses, we hypothesized that CB2Rs expressed in microglia and dopamine neurons modulate the behavioral responses in CB2R conditional knockout (cKO) mice. The data from the studies revealed that CB2Rs in microglia and dopamine neurons are implicated in neuro-immune modulatory effects of CB2Rs in mouse models of CNS disorders. In modeling depression in the mouse models, it is important to note that there are different types of depression with co-morbidities with mental and neurological disorders. There are limitations, and, to be useful, an animal model does not have to be a perfect replication of the disorder in terms of the predictive, face, and construct validity. For example, major depression is characterized by a depressed mood and anhedonia—a lack of pleasure—including other symptoms like reduced appetite and libido and disturbed sleep. While anhedonia can be modeled in rodent studies, a depressed mood is difficult to characterize. However, preclinical and clinical studies showed that the ECS is responsible for modulating depression, with reports that there is a negative correlation between the ECS signaling and depression [76]. The withdrawal of rimonabant, a CB1R antagonist, used for the treatment of obesity due to the risk of suicide and depression, has received increased interest as regards the role of CB1R in affecting mood and affective behavior. Some studies have investigated the role of CB1Rs in depression with more basic and clinical components, and factors like stress, the interaction with the endocrine system, and the use of cannabis and depression, with revealing outcomes. On the other hand, the involvement of the CB2R in affective disturbances has not received much attention [49]. Previous studies indicate that the overexpression of CB2R provoked a depression-like response. CB2xP mice (overexpressing CB2R) reduced the immobility time in acute models of depression (tail suspension and novelty-identification and feeding tests) [72]. Mutant (*Cnr2−/−* mice) developed a depressive-like behavior in the tail suspension test [56]. In line with this, the results found with the dopamine neuron cell-type-specific CB2R conditional knockout (DAT-*Cnr2*) mice demonstrated a depressive-like behavior indicated by an enhanced immobility time in the tail suspension and forced swimming tests [12]. However, a pharmacologic study using GW405833, a CB2R agonist, demonstrated that the acute administration of the agonist did not alter the immobility time in the forced swim test [77].

It is important to point out the validity and the mixed and conflicting responses in the use of some of these animal models of depression, that have been questioned in the scientific community. Nevertheless, the CMS is a commonly used animal model of depression, with similar behavioral and physiological effects observed in clinical settings in patients with depressive disorders [78,79,80]. The involvement of CB2R in depression was demonstrated in a study using mice subjected to CMS. The results showed that CB2R protein levels measured by immunoblotting in a whole brain extract were enhanced in mice subjected to CMS for a period of 4 weeks [17,45]. However, another study showed a reduction in the level of CB2R mRNA in the hippocampus of mice subjected to CMS for 7–8 weeks, compared with non-stressed controls [72]. The contradiction might be due to differences in the time of exposure to CMS and variations in the brain regions used to measure the level of CB2R mRNA. The CMS protocol is characterized by a reduced intake of sucrose, used as a measure of anhedonia-like response, a hallmark of depression. Studies using transgenic mice that over-express the CB2R showed that there is no change in sucrose consumption and immobility time in mice subjected to a tail suspension test, respectively [72]. However, in another study done to evaluate the effect of CB2R ligands on sucrose consumption and the daily injection of JWH-015, a CB2R agonist, or AM630, a CB2R antagonist, did not alter the CMS-induced decreases in sucrose consumption [16,17]. Taken together, the results from the preclinical studies indicate the possible behavioral and molecular role played by CB2Rs in CMS. However, there is still a limited literature on the role of CB2Rs in depression in humans. In one study conducted on Japanese patients with depression, the authors found that there is a high incidence of the Q63R polymorphism of the CB2R gene [16]. Furthermore, a reduction in the expression of the CB2R gene was documented in brain regions involved in regulating emotionality like the amygdala and the prefrontal cortex in a postmortem study performed in suicide patients [81] and in patients with major depression [82], indicating the role that CB2Rs could play in depression and the potential of these receptors as a new target for the treatment of depressive disorders, but this hypothesis requires additional modeling and clinical investigations.

#### 4.1.3. Schizophrenia and CB2R-ECS

Schizophrenia is a type of psychotic disorder that is characterized by thinking disturbances, hallucinations, and delusions associated with genetic and epigenetic risk factors. Disturbances in the regulation of the ECS with altered levels of eCBs were associated with the development of schizophrenia. Indeed, the densities of eCB receptors and the levels of eCBs have been suggested as possible biomarkers in neuropsychiatric disorders [75]. Thus, the hypothesis on the role of cannabinoid self-medication, the early exposure to cannabis, and the onset of psychosis in vulnerable adolescents has fueled a research interest on the changes in the ECS in neuropsychiatric disorders. Recently, there is also an increasing interest in the use of herbal and synthetic eCBs for the symptomatic management of schizophrenia [83]. Studies using animal models of schizophrenia indicated the involvement of CB2R in schizophrenia. Early maternal deprivation in rodents is a model for neurodevelopmental stress, and there are some reports indicating that maternal deprivation affects the ECS and that these changes may account for the proposed schizophrenia-like phenotype [84,85]. A study to evaluate the expression of CB1R and CB2R in the hippocampus showed that maternal deprivation induced a significant increase in CB2R immunoreactivity in the hippocampal areas, indicating the possible involvement of CB2Rs in neurodevelopmental mental illnesses such as schizophrenia [86]. The pre-pulse inhibition (PPI) test is one of the tests used to assess attention deficits associated with schizophrenia [13]. Methamphetamine and MK-801, a non-competitive N-methyl-D-aspartate (NMDA) glutamatergic receptor antagonist, are two widely used animal models of schizophrenia that mimic the effects observed in schizophrenic individuals [87]. Studies showed that AM630, a CB2R antagonist, did not affect PPI on its own, but it did enhance the MK-801- and metamphetamine-induced decrease in PPI and the increase in locomotor activity [88]. Another study found that the administration of JWH015 in doses of 1, 3, and 10 mg/kg enhanced the PPI impairment caused by MK-801 [89]. CB2R’s role in schizophrenia was investigated in genetically engineered mice. It was reported that *Cnr2*−/− mice were more susceptible to hyperlocomotion caused by acute cocaine exposure, as well as PPI and cognitive changes. Furthermore, in these mutant *Cnr2*−/− mice, treatment with the antipsychotic risperidone normalized the PPI readings [56,90]. Interestingly another study showed that CB2R is required for the antipsychotic effects of acetylcholine muscarinic 4 (M4) receptor agonists like UV0467154 [91]. This study found that stimulating the M4 receptors increased the release of eCBs while inhibiting dopamine release, implying that M4 agonists have a role in antipsychotic efficacy. As a result, inhibiting CB2Rs with the antagonist AM630 would prevent the antipsychotic effects of M4 agonists. Preclinical studies also showed that cannabidiol (CBD), another major phytocannabinoid, which is devoid of intoxicating effects, has antipsychotic properties [92,93,94,95] mediated by the activation of CB2Rs [96,97,98]. It should be pointed out that CBD is a non-intoxicating rather than a non-psychoactive constituent in the cannabis plant. According to the findings, CB2Rs play a significant role in the potentiation of antipsychotic medication effects. Clinical studies have also shown that CB2R is involved in schizophrenia. *CNR2* gene expression was shown to be significantly reduced in the peripheral mononuclear blood cells of patients with schizophrenia [99]. In Japanese schizophrenia patients, the *CNR2* polymorphisms rs12744386 and rs2501432 were found to be significant (Ishiguro et al., 2010), whereas the *CNR2* polymorphisms rs2501432 and rs22229579 were found to be significant in the Han ethnic population of China [100]. These findings imply that CB2R plays a critical role in modulating psychotic symptoms. However, further controlled clinical trials are needed to determine the role of CB2Rs in the treatment or as an adjunctive strategy in schizophrenia.

#### 4.1.4. CB2R-ECS in Addiction

The global drive and use of cannabis and cannabinoids for recreation and medicine requires a focus on the emerging complex eCBome in reward processing and drug addiction in emerging cannabis use disorders (CUDs). With the emergence of the COVID-19 pandemic, and the devastating global consequences, drug addiction and especially the opioid epidemic and overdoses and death, increased alcohol consumption, and obesity are major concerns [66]. The discovery of this previously unknown but ubiquitous ECS, and the fact that CBRs are encoded in the human genome on chromosomes 1 and 6, have led to the changing landscape on cannabinoid research and provided scientific evidence on the potential therapeutic benefits of targeting the ECS. There are still gaps in our understanding, but the discovery of the expression of CB2Rs in brain regions involved in drug addiction, such as the ventral tegmental area (VTA), nucleus accumbens (NAc), amygdala, and hippocampus unraveled and encouraged the investigation of the role of CB2Rs in drug addiction [101]. Therefore, the increasing interest and attention on the CB2Rs as a target for the treatment of drug addiction is due in part to its neuro-immune functioning associated with the reward pathway. Pharmacologic and genetic studies showed that CB2Rs are involved in the effects of psychostimulants like cocaine, amphetamines, and methamphetamines. CB2R is involved in cocaine motor sensitization in studies employing genetically modified mice. Mice overexpressing CB2R in the (CB2xP) showed reduced hyperlocomotor effects in response to acute cocaine administration and were less susceptible to the motor sensitization after repeated cocaine administration [102]. Interestingly, mice lacking the CB2R (*Cnr2*−/−), which have increased sensitivity to cocaine-induced hyperlocomotion, had the opposite effect [56].

A study aiming to evaluate the effects of psychostimulants in dopamine neuron CB2R conditional knockout (DAT-*Cnr2* cKO) mice on locomotor activity showed that mice had increased cocaine-, amphetamine-, and methamphetamine-induced hyperactivity, without psychostimulant-induced sensitization when compared to wild-type controls. In addition, the authors reported that cocaine, amphetamines, and methamphetamines produced a robust conditioned place preference (CPP) in both DAT-*Cnr2* cKO and wild-type mice [55]. In stark contrast, studies using JWH133 revealed a systemic and local administration of JWH133 into the NAc-blocked cocaine hyperlocomotion in mice [41,103]. In line with this, a study revealed that JWH133 inhibited cocaine and nicotine-induced CPP in wild-type mice [55]. However, in another investigation, AM630 was found to have no effect on the locomotor effects after an acute or repeated cocaine administration in rats [103,104]. These inconsistencies could be explained by differences in animal species and the methods utilized. Since cocaine produced place aversion rather than place preference in CB2xP animals, and these transgenic mice showed an impairment in the acquisition of cocaine self-administration [102], CB2Rs appear to be implicated in the reinforcing features of cocaine. Evidence also showed that the intranasal or intra-accumbens administration of JWH133 prevented intravenous cocaine self-administration [41]. The involvement of CB2Rs in a model of addictive behavior has been demonstrated by the study using the natural CB2R agonist β-caryophyllene (BCP). The results showed that BCP attenuated methamphetamine self-administration in rats, which was partially blocked by AM630 [105]. The study also revealed that the deletion of CB2Rs blocked a low-dose BCP-induced reduction in methamphetamine self-administration. There are also investigations on the role of CB2Rs in the modulation of alcohol consumption and reward. The injection of BCP lowered the ethanol CPP and ethanol intake in mice, and increased the voluntary ethanol consumption in the two-bottle paradigm and provided an incentive to drink in the oral ethanol self-administration in the mutant *Cnr2*−/− mice [106,107]. In addition, a sub-chronic injection of JWH015 enhanced the alcohol intake in mice that had previously been exposed to chronic stress, with no impact in mice that had not been exposed to chronic stress [64,108]. A recent study also demonstrated that alcohol exposure during adolescence induced increases in the amygdala expression of the CB2Rs in rats [109]. A substantial incidence of the single nucleotide polymorphism R63Q in the *CNR2* gene locus was observed in a cohort of Japanese alcoholic patients in clinical investigations [64]. This single nucleotide polymorphism causes a missense mutation in the first intracellular domain, which reduces the physiological responsiveness to CB2R ligands [101]. More recently, Navarrete et al. extensively complied a significant appraisal of type-2 cannabinoid receptor involvement in the treatment of substance use disorders (SUDs), including alcohol and psychostimulants (cocaine and nicotine), with an emphasis on the CB2R species differences in the effects of the SUDs covered (198). They concluded that when combined with existing options, the development of pharmacotherapeutic agents targeting the CB2Rs might offer novel approaches [110]. It is noteworthy that preclinical investigations show that CB2R ligands can modulate the reward system and increase the understanding of neuroinflammation in CNS disorders. All the above data support the role of CB2Rs in modulating the addictive effects of alcohol and different psychostimulants, indicating that CB2Rs might be targeted in the treatment of drug addiction. Therefore, there is now an increased focus and interest in CB2Rs in drug and alcohol addiction, with increasing reports indicating their functional expression not only in reward circuits but also on the CB2R-neuro-immune crosstalk and signaling in neuropsychiatric disorders.

#### 4.1.5. CB2R-ECS in Autism Spectrum Disorder (ASD)

With the recent advances in cannabis and cannabinoid research, along with the increased acceptance of cannabis medicine, cannabinoids have been approved in childhood disorders including epilepsy, as discussed below, and in the Autism spectrum disorder (ASD). ASD has an early childhood onset and a lifelong progression. It is characterized by poor social skills, deficits in communication, as well as stereotypic behaviors [111]. The ECS plays a significant role in ASD. A review on the role of the ECS in autism found that endocannabinoid signaling plays a key role in many human health and disease conditions of the CNS [112]. However, there is limited literature, pre-clinical and clinical, on the role of CB2Rs in ASD. CB2Rs were also identified as a possible target for autism in preliminary research. Indeed, in our earlier studies, in the cerebellum of BTBR T + tF/J, a mouse model of ASD, we reported an increased level of CB2A mRNA expression, but not of the CB2BR, gene-transcript isoforms [15]. A controlled study of ASD conducted on autistic children showed that gene expression for *CNR2* but not *CNR1* was up-regulated in peripheral blood mononuclear cells (PBMCs). The result indicated that *CNR2* gene expression was significantly higher in individuals with ASD compared to controls. In addition, the gene expression for one of the enzymes responsible for synthesis of the endocannabinoid anandamide (NAPE-PLD) was significantly lower in individuals with ASD. This could have caused an increase in CB2Rs resulting from a decrease in AEA synthesis, which may be indicative of a decreased ECS tone in ASD [113]. While there is a multifactorial and putative link of the ECS to ASDs, there are no solid biomarkers. Others are increasingly convinced that a medical cannabis treatment may yield cannabis-responsive biomarkers and that ECS may be altered in ASDs. However, our investigations using BTBR T + tF/J and other preclinical mouse models rely on the characterization of behavioral alterations [15]. There is a dearth of basic and clinical research, although our previous studies may support the development of cannabinoids targeting the ECS components. There is support for such a view, as the ECS components provide CB2R neuro-immuno- modulatory targets for ASDs [114]. It is important that more basic and clinical research be conducted, and trials may unravel the etiology of ASDs, which involve not only multigenetic risk factors but also environmental neurodevelopment factors in utero. With the increasing knowledge on the alteration of the ECS in ASDs, the initial relief of ASD symptoms requires trials to ascertain the use of cannabis and cannabinoids modulating the ECS in ASDs.

#### 4.1.6. CB2R-ECS in Eating Disorders

It is well known that the ECS modulates appetite—a basis for adjunctive cannabinoids use to counter AIDs cachexia—and regulates the metabolism of glucose in the pancreas and liver. “Diabesity” refers to the relationship between diabetes and obesity. Even though the leading cause of this metabolic disorder is a poor life style, diabesity is in part linked with ECS signaling in adiposity and metabolism. This indicates that the dysregulation of the gut ECS and altered levels of eCBs are associated with obesity and metabolic syndrome. The physiological roles of the ECS in the gastrointestinal tract (GIT), adipogenesis, and lipogenesis provide an increasing link in the dysfunction of the ECS with obesity, metabolic syndrome, and other disturbances in the periphery. The most common eating disorders are anorexia nervosa (AN) and bulimia nervosa (BN), presenting abnormal eating behaviors that generally result in severe food restriction with episodes of binge eating and vomiting, without significant changes of body weight in BN and a dramatic loss of body weight in AN [111]. The hypothalamic eCBs have been shown to modulate eating behavior in animals and humans, thus indicating the role of the ECS in the pathophysiology of eating disorders [115,116]. In a study conducted on 20 women with AN, 23 women with BN, and 26 healthy women, the levels of CB1R and CB2R expression were examined. The study found no differences in CB2R mRNA levels in the blood of AN and BN patients when compared to controls, with no significant difference between the two groups [117,118]. So far, there is only one study on human genetic association to identify whether the *CNR2* gene is involved in eating disorders [119]. A total of 204 individuals with eating disorders (94 AN and 111 BN) and 1876 healthy Japanese volunteers participated in the study. A nonsynonymous *CNR2* polymorphism was found to be linked to both AN and BN. The R allele is substantially more prevalent in people with eating disorders than in controls, according to the findings. Furthermore, there was no change in allele frequency between patients with AN and BN when they were separated. The studies demonstrate a link between CB2Rs and eating disorders when taken together, but more research is needed to corroborate these findings.

**Table 1 ijms-23-00975-t001:** CB2 cannabinoid receptors in neuropsychiatric disorders.

Disorder	Study Type	Model/Paradigm	CB2R Manipulation	Outcome	Reference
Anxiety/Anxiety-like behavior	Preclinical	Chronic mild stress/ Elevated plus maze	JWH015	Induced angiogenesis	[18,42]
		Marble burying	GW405833	Induced anxiolysis	[73]
		Chronic mild stress/ Zero maze	JWH015	Reduced anxiety like behavior	[74]
			AM630	Increased anxiety-like behavior	
		Light-dark box, Elevated plus maze	AM630	Induced anxiogenesis and anxiolysis after acute and chronic administrations	[53]
			CB2xP mice	No response to anxiogenic-like stimuli	
		Light-dark box, Elevated plus maze	*Cnr2−/−* mice	Enhanced anxiety-like behavior	[56]
		Light-dark box, Elevated plus maze, Forced swim and tail suspension	DAT-*Cnr2* cKO mice, JWH133	Increased anxiety-like behavior	[12]
	Clinical	Children with anxiety		Significant relationship between rs2070956 polymorphism and treatment outcome	[75]
Depression	Preclinical	Tail suspension, novelty-suppressed feeding test, Chronic mild stress	CB2xP mice	Reduced immobility time	[72]
		Light-dark box, Elevated plus maze	*Cnr2−/−* mice	Developed depressive-like behavior	[56]
		Forced swim and tail suspension	DAT-*Cnr2* cKO mice, JWH133	Increased immobility time	[12]
		Forced swim test	GW405833	No change in time spent immobile	[77]
		Chronic mild stress	JWH015	Enhanced CB2R protein level	[17,45]
		Chronic mild stress	CB2xP mice	Reduced CB2R mRNA	[72]
	Clinical	Japanese depressive patients		High incidence of Q63R polymorphism of CB2R	[16]
		Postmortem study		Reduced expression of CB2R gene	[81,82]
Schizophrenia	Preclinical	Early maternal deprivation in rats		Increased CB2R immunoreactivity in the hippocampus	[86]
		MK-801, methamphetamine	AM630	Did not affect pre-pulse inhibition alone but enhanced MK-801 or methamphetamine induced effect	[119]
		MK-801	JWH015	Enhanced pre-pulse impairment caused by MK-801	[89]
		Acoustic pre-pulse inhibition	*Cnr2−/−* mice	Decreased pre-pulse inhibition	[56,90]
		MK-801, Pre-pulse inhibition	*Cnr2−/−* mice, AM630	AM630 inhibited the ability of VU0467154 to reverse disruption of pre-pulse inhibition	[91]
		Methylazoxymethanol acetate	Cannabidiol	Prevent schizophrenia-like deficits	[92,93]
	Clinical	Schizophrenic patients		Significant reduction in the expression of *CNR2* gene in peripheral mononuclear blood cells	[99]
		Japanese schizophrenia patients		Significant association between *CNR2* polymorphisms rs12744386 and rs2501432	[119]
		Chinese schizophrenic patients		Significant association between *CNR2* polymorphisms rs22229579 and rs2501432	[100]
Addiction	Preclinical	Self-administration, conditioned place preference	CB2xP	Decreased cocaine motor sensitization and self-administration	[102]
		Open field test	*Cnr2−/−* mice	Enhanced cocaine motor sensitization	[56]
		Open field, conditioned place preference	DAT-*Cnr2* cKO mice	Increased psychostimulant induced motor sensitization and conditioned place preference	[55]
			JWH133	JWH133 inhibited cocaine and nicotine induced conditioned place preference	
		Open field, conditioned place preference	*Cnr2−/−* mice, JWH133	JWH133 blocked cocaine locomotion and self-administration	[41,103]
		Drug self-administration under fixed and progressive ration	β-caryophyllene	Attenuated methamphetamine self-administration	[105]
			*Cnr2−/−* mice	Blockage of β-caryophyllene induced reduction in methamphetamine self-administration	
		Alcohol consumption and place preference	*Cnr2−/−* mice	Enhanced ethanol conditioned place preference	[106]
			β-caryophyllene	Decreased ethanol consumption and preference	[107]
		Chronic mild stress	JWH133	Enhanced alcohol intake	[64,108]
		Alcohol consumption		Increased amygdalar expression of CB2Rs	[109]
	Clinical	Japanese alcoholic patients		Single nucleotide polymorphism R63Q in *CNR2* gene	[64]
Autism Spectrum Disorder	Preclinical	BTBR T + tF/J		Increased level of CB2AR mRNA	[15]
	Clinical	Children with autism spectrum disorder		Up-regulation of expression of *CNR2* gene in peripheral blood mononuclear cells	[113]
Eating Disorders	Clinical	Patients with eating disorders		No change in CB2R mRNA level in blood of the subjects	[117,118]
		Japanese patients with eating disorder		Link between *CNR2* gene polymorphism and eating disorder	[119]

### 4.2. Potential Role of CB2Rs in Neurologic and Neurodegenerative Disorders

#### 4.2.1. CB2R-ECS in Alzheimer’s Disease (AD)

Because of the neuroprotective role of CB2Rs in CNS disorders with neuroinflammatory biomarkers, there has been interest and focus on the potential of targeting CB2Rs in Alzheimer’s disease (AD) in a number of animal models of AD. AD is characterized by the abnormal accumulation of β-amyloid (Aβ) in senile plaques in the brain, which causes cognitive impairment, memory loss, and behavioral changes due to neurodegeneration and inflammation [120]. Studies demonstrate that the ECBs have been shown to decrease Aβ-induced microglia activation and neuroinflammation [121,122,123]. Recently, CB2Rs have attracted attention in AD investigation because of their expression in immune cells and their enhanced expression during inflammation. Studies showed that there is increased expression of CB2Rs in brain tissue in AD patients and mouse models expressing pathogenic variants of the amyloid precursor protein (APP) [121,124,125]. Previous research found that the genetic deletion of CB2Rs increased Iba1 staining and exacerbated soluble A42 and plaque deposition [126], implying that CB2Rs play a key role in preventing amyloid plaque pathology in AD [127]. In mice, the administration of JWH-133 improved cognitive impairment, inhibited neuroinflammation and oxidative stress, decreased tau hyperphosphorylation, induced vasodilation [128,129], and enhanced glucose uptake, indicating that CB2R agonists could be used as nootropics [130]. A recent study assessing the role of AEA analog-N-linoleyltyrosine (NITyr) in APP/PS1 mice mimicking the AD model showed that NITyr protected neurons against Aβ injury, which is mainly mediated by the CB2Rs [131]. Interestingly Rivas-Santisteban et al. [132] demonstrated that CB2R activation blunted NMDA receptor-mediated signaling in primary hippocampal neurons from a APP_Sw/Ind_ mice model of AD, suggesting a role of CB2Rs in AD pathogenesis. In addition, in AD model mice, CB2R activation by JWH-015 was found to improve new object recognition abilities while also regulating microglia-mediated neuroinflammation and dendritic complexity in a region-specific manner (Li et al., 2019) [133]. A parenteral administration of 1-((3-benzyl-3-methyl-2,3-dihydro-1-benzofuran-6-yl) carbonyl) piperidine (MDA7), a novel selective CB2R agonist, inhibited the activation of microglial cells and astrocytes, decreased the level of expression of CB2R, enhanced the clearance of Aβ, and improved cognition and memory in rodent AD models [134,135].

In various in vitro and in vivo AD models, CB2R activation reduced the levels of neurotoxic factors and pro- inflammatory mediators produced by reactive astrocytes and microglial cells, stimulated microglial proliferation and migration, and decreased Aβ levels [136,137,138]. Despite a large body of evidence for the involvement of CB2Rs in reducing and processing Aβ in a mouse model of AD, it is difficult to determine the therapeutic value of cannabis-based medicines in AD [139]. There are also reports that show that mice with CB2R deficiency showed a reduction in microglial cells and macrophages, reduced expression levels of brain pro-inflammatory cytokines, diminished concentrations of soluble Aβ40/42, and improved cognitive and learning deficits [121]. In a few clinical studies, there are contradictory reports on the potential of the CB2R ligand in AD, and further research is required to determine the potential role of CB2Rs in humans and in animal models of AD.

#### 4.2.2. CB2R-ECS in Parkinson’s Disease (PD)

Earlier studies indicated changes in the ECS particularly with the interaction of the dopaminergic system and eCBs in basal ganglia associated with movement disturbances, one of the hallmarks of Parkinson’s disease (PD). PD is a movement disorder characterized by the degeneration of dopaminergic neurons, that results in motor dysfunction [140,141]. Disease progression is characterized by inflammation through the activation of microglia [142] and an increase in cytokines [143,144]. Accumulated evidence suggests that there is a strong potential for the ECS to provide neuroprotection against acute or chronic neurodegenerative disorders [145,146,147,148,149]. CB2R levels were significantly elevated in animal models of PD and postmortem studies of PD patients, and this increase correlated significantly with an increase in microglial activation, indicating the possible role of CB2Rs in PD [61,150]. Studies showed a down-regulation of CB2Rs in the substantia nigra and hippocampus three weeks after 1-methyl-4-phenyl-1,2,3,6-tetrahydropyridine (MPTP) injection in mice with MPTP-induced parkinsonian syndrome. In addition, AM1241, the selective CB2R agonist, has been shown to regenerate DA neurons after the neurotoxic effect of MPTP treatment [145]. Furthermore, mice lacking the CB2R showed an enhanced activation of microglial cells and a much more intense deterioration of tyrosine hydroxylase (TH)-containing nigral neurons in animal models of PD [150], which supported the potential neuroprotective role of CB2Rs.

In animal models of PD, Δ^9^-THCV, a CB2R agonist, reduced the motor inhibition caused by 6-hydroxydopamine (6-OHDA) and the loss of TH–positive neurons caused by a 6-OHDA lesion in the substantia nigra after acute and chronic administrations, respectively. However, CB2Rs were poorly up-regulated in the rats’ substantia nigra in response to 6-OHDA. By contrast, the substantia nigra of mice that had been injected with lipopolysaccharide (LPS) exhibited a greater up-regulation of CB2Rs. The authors suggest that Δ^9^-THCV caused the preservation of TH–positive neurons, probably through the involvement of CB2Rs [151]. In another study the protective effect of JWH-015 against MPTP-induced nigrostriatal degeneration and the suppression of microglial activation/infiltration through the activation of CB2Rs [152] were implicated. In addition, there are reports showing that BCP reduced oxidative stress and neuroinflammation, blocked gliosis and pro-inflammatory cytokine release, and lowered nigrostriatal degeneration in a rotenone (ROT)-induced animal model of PD [153]. Like the pre-clinical studies, clinical studies revealed that the expression of CB2Rs in different brain cells is up-regulated in PD patients. One study demonstrated that CB2Rs are elevated in microglial cells recruited and activated at lesioned sites in the substantia nigra of PD patients compared to control subjects [150]. In another study, the authors observed that CB2R was located in TH-containing neurons in the substantia nigra at significantly lower levels in PD patients compared to controls [154]. Thus, CB2Rs may be a promising pharmaceutical target for alleviating parkinsonian symptoms and slowing disease development in Parkinson’s disease.

#### 4.2.3. CB2R-ECS in Huntington’s Disease

Huntington’s disease (HD) is a neurodegenerative disease caused by the expansion of the CAG triplet repeat (cytosine-adenine-guanine) in the gene encoding the protein huntingtin (*Htt*), which leads to cognitive decline and abnormal motor movements (chorea) [1,155]. Currently there is no effective treatment for HD, and the need for new therapeutic targets is increasing [1]. Because the ECS is abundant in the basal ganglia, the activation or inhibition of the ECS signaling pathway may have a major impact on motor responses [156], and CB2R is emerging as a new therapeutic target for the treatment and early diagnosis of HD [157]. In the transgenic R6/2, CAG repeat length Huntington chorea mouse model, CB2R expression was shown to be elevated in the hippocampus, brain, striatum, and cerebellum [158]. Studies also found that mice lacking CB2Rs were more sensitive to malonate than the control group. In addition, CB2R-defficient mice exhibited an enhanced onset of motor deficits and increased severity [159,160]. The expression of CB2R was increased in striatal microglia in a transgenic mouse model of HD and in patients. In addition, the genetic ablation of CB2R exacerbated HD, and the administration of CB2R-selective agonists reduced the striatal neurodegeneration through microglial activation [161]. Taken together, CB2Rs might be a potential target and compounds that selectively activate CB2R might be utilized as a potential therapeutic agent in the treatment of HD.

#### 4.2.4. CB2R-ECS in Multiple Sclerosis

Multiple sclerosis (MS) is an autoimmune disorder characterized by the inflammation, neurodegeneration, and demyelination of neurons with no effective therapy [162], and studies showed the role of CB2Rs in inflammatory conditions associated with MS [157,163]. Microglial activation was linked to CB2R overexpression in an experimental autoimmune encephalomyelitis (EAE) animal model of MS [164]. HU-308, the CB2R-selective agonist, also dramatically reduced the EAE symptoms, axonal loss, and microglial activation when administered orally [165]. O-1966, a CB2R agonist, was also found to reduce immune cell invasion, diminish white cell rolling and adhesion to cerebral microvessels, and improve the neurologic function following an insult on the EAE progression [166]. Additional research conducted using EAE animal models demonstrated that BCP significantly inhibits microglial cells, CD4+ and CD8+ T lymphocytes, and cytokines, diminished axonal demyelination, and modulated the Th1/Treg immune balance through the activation of CB2Rs [167]. Maresz et al. [164] also reported that the CB2R expression by encephalitogenic T cells reduced the EAE-associated inflammation. Furthermore, during EAE, CB2R-deficient T cells in the CNS showed reduced apoptosis, increased proliferation, and increased production of inflammatory cytokines, resulting in severe clinical illness. The Theiler murine encephalomyelitis virus-induced demyelinating disease (TMEV- IDD), which mimics the central neuronal demyelination that occurs in MS, is another animal model of MS [168]. In TMEV-IDD mice, the CB1R and CB2R agonists showed enhanced clinical benefits through immunomodulatory and anti-inflammatory mechanisms [169,170], which is another evidence for the role of CB2Rs in MS. Postmortem and clinical studies also showed the involvement of CB2Rs in MS. An association between the CB2R rs35761398 (Q63R) polymorphism and MS was found in a study conducted on a total of 100 Iranian MS patients and 100 healthy controls [171]. Evidence suggest that MS patients showed a significantly elevated CB2 expression in B cells, but not in T or NK cells [172]. Yiangou et al. [173] indicated a higher level of microglia cells in human postmortem spinal cord specimens. Evidence revealed that T lymphocytes, astrocytes, and both perivascular and reactive microglial cells were also observed to express CB2Rs in postmortem brain tissues from MS donors [174]. These findings suggest that CB2Rs play a neuroprotective function in MS pathology, and targeting CB2Rs could aid in the management of symptoms and neurologic issues in MS patients.

#### 4.2.5. CB2R-ECS in Amyotrophic Lateral Sclerosis

Amyotrophic lateral sclerosis (ALS) is a degenerative disease affecting the cortex, the brain stem, and the spinal cord motor neurons. The deterioration of both the upper and lower motor neurons is the main feature of disease progression [121,175]. The involvement of CB2Rs in ALS has been demonstrated by different studies. A reduction in motor neuron degeneration and the preservation of the motor function in ALS was observed after a selective activation of CB2Rs in animals [176,177,178]. TDP-43 transgenic mice and postmortem specimen studies showed that there is an up-regulation of CB2Rs in activated microglia cells [173,179]. Studies also showed that the survival interval after ALS onset was increased by 56% after the administration of the CB2R agonist AM1241, initiated at symptom onset [176]. In addition, Kim et al. [177] reported a reduction in the signs of disease progression when AM1241 was administered after the onset of signs in an ALS mouse model (hSOD1(G93A) transgenic mice. Another study done to evaluate the neuroprotective effect of the phytocannabinoid-based medicine Sativex^®^ using SOD1G93A mutant mice, an experimental model of ALS, showed a significant increase of CB2Rs [180]. All these evidence indicate that CB2R plays a role in preventing the progression of ALS. Therefore, CB2Rs might be considered a promising target for therapeutic approaches in ALS.

#### 4.2.6. CB2R-ECS in Epilepsy

Epilepsy is a common neurological illness, affecting more than 70 million individuals worldwide. The currently available anti-epileptic drugs are unable to control epileptic seizures in the majority of patients [181], and hence there is a need for new therapeutic targets for effective anti-epileptic drugs. There is evidence on the role of the ECS in epilepsy [182,183], and studies showed that CBD was effective in the treatment of epileptic seizures in preclinical [184] and clinical studies [185,186]. The role of CB2Rs in epileptic seizures has been documented in other studies [33,187]. A study showed that the selective CB2R antagonist AM630 inhibited a palmitoyl ethanolamide (PEA)-induced increase in the latency of seizure initiation and reduced the duration of seizures in an acute pentylenetetrazol (PTZ) rat seizure model [188]. In addition, the administration of BCP was found to improve seizure activity in a mouse model [189]. Despite the evidence showing the role of CB2Rs in controlling seizures in animals, few studies found a possible association between the activation of CB2Rs and the control of seizures. In a study using a HU-308, a CB2R selective agonist, HU-308 did not show an antiseizure effect, and AM630 did increase seizure severity [190]. Moreover, the CB2R agonist AM1241 increased seizure intensity in a PTZ model [191], and AM630 and SR144528 can increase seizure susceptibility [190]. Studies using genetically manipulated mice revealed that CB2R knockout mice show an enhanced epileptic susceptibility, and a reduction in CB2R activity was associated with increased susceptibility [192]. Taken together, accumulated evidence support the involvement of CB2Rs in the pathophysiology of epileptic seizures. The enhanced expression of CB2Rs in the brain during various disease states including epilepsy make them an important target for neuroprotection, and hence CB2Rs might serve as a possible target for the treatment of epilepsy.

#### 4.2.7. CB2R-ECS in Traumatic Brain Injury

Traumatic brain injury (TBI) is caused by a mechanical injury of the brain that results in the formation of a hematoma that ultimately leads to long-term complications and death [193]. Evidence show the involvement of CB2Rs in modulating the pathophysiology of TBI. Studies using the CB2R agonist, O-1966, in mice with TBI indicated that the administration of O-1966 attenuated the disruption of the blood–brain barrier and neuronal degeneration [194], and induced acute neuroprotective effects [195], supporting a role of CB2Rs in the management of TBI. There are also reports showing an increase in the expression of CB2Rs in mice with TBI associated with edema neurologic deficits. In that study, the authors found a positive correlation between the expression of CB2Rs and TBI [59]. In another study aimed at evaluating the potential effects of the CB2R agonists, HU-910 and HU-914, in the pathophysiology of TBI in mice, Magid et al. [196] demonstrated an enhanced neuroprotection and neurobehavioral recovery. In a recent study, the selective CB2R agonist, JWH133, protected white matter injury via PERK signaling in a rat model of traumatic brain injury [197].

**Table 2 ijms-23-00975-t002:** CB2 cannabinoid receptors in neurological and neurodegenerative disorders.

Disorder	Study Type	Model/Paradigm	CB2R Manipulation	Outcome	Reference
Alzheimer’s Disease	Preclinical	AβPP/PS1 transgenic mice	JWH133	Reduced tau hyperphosphorylation, induced vasodilation	[128,129]
			JWH133	Enhanced brain glucose uptake	[130]
			N-linoleyltyrosine	Protected neurons against Aβ injury	[131]
		APPSw/Ind	JWH133	CB2R activation blocks NMDA signaling in activated microglia	[132]
		APP/PS1 mice	JWH015	Improved novel object recognition, regulation in microglia-mediated neuroinflammation and dendritic complexity	[133]
		Double transgenic APP/PS1 mice	1-((3-benzyl-3-methyl-2,3-dihydro-1-benzofuran-6-yl) carbonyl) piperidine (MDA7)	Inhibited microglia activation, enhanced clearance of Aβ and decrease level of CB2R expression	[134,135]
Parkinson’s Disease	Preclinical	MPTP model	AM1241	Prevented neurodegeneration	[145]
		Lipopolysaccharide (LPS) model	*Cnr2−/−*	Enhanced activation of microglia	[150]
		6-hydroxydopamine (6-OHDA)	Δ^9^-THCV	Reduced motor inhibition and loss of TH-positive neurons caused by 6-OHDA, reduced CB2Rs up-regulation	[151]
		Lipopolysaccharide (LPS) model		Exhibited a greater up-regulation of CB2Rs	
		MPTP model	JWH015	Protected MPTP induced neurodegeneration and suppress microglia activation	[152]
		Rotenone (ROT) animal model	β-caryophyllene	Reduced oxidative stress and neuroinflammation	[153]
	Clinical	PD patients		Elevated CB2Rs in microglia cells in the substantia nigra	[154]
Huntington’s Disease	Preclinical	Transgenic R6/2 mouse model		Elevated CB2R expression in the hippocampus, striatum and cerebellum	[158]
		Malonate rat model	*Cnr2−/−*	CB2R activation protected striatal neuron degeneration	[159]
		R6/2 mice	*Cnr2−/−*	CB2R agonist suppresses motor deficits, synapse loss, and CNS inflammation	[160]
		R6/2 mice	*Cnr2−/−*	Increased CB2R expression in striatal microglia	[161]
Multiple Sclerosis	Preclinical	Experimental autoimmune encephalomyelitis (EAE)		Enhanced CB2R expression	[164]
			HU308	Reduced symptoms, axonal loss and microglia activation	[165]
			O-1966	Reduced immune cell invasion and improved neurologic functions	[166]
			β-caryophyllene	Inhibited activation of immune cells and diminished axonal demyelination	[167]
		Theiler’s murine encephalomyelitis virus model	JWH015	Improved the neurological deficits in a long-lasting way, induced anti-inflammatory response	[169,170]
	Clinical	Iranian multiple sclerosis patients		Significant association between Q63R gene polymorphism and multiple sclerosis	[171]
		Multiple sclerosis patients		Elevated CB2R expression in B cells	[172]
		Human postmortem specimen		Enhanced microglia cells	[173]
		Postmortem brain tissue		Enhanced expression of CB2Rs	[174]
Amyotrophic Lateral Sclerosis	Preclinical	TDP-43 transgenic mice		Up-regulation of CB2Rs	[179]
		G93A-SOD1 mice	AM1241	Slowed motor neuron degeneration and preserved motor function	[176]
				Delayed disease progression	[177]
			Sativex^®^	Increased level of CB2Rs	[180]
	Clinical	Postmortem specimen		Up-regulation of CB2R	[173]
Epilepsy	Preclinical	Palmitoylethanolamide (PEA)	AM630	Blocked almitoylethanolamide induced seizure	[188]
		Pilocarpine, pentylenetetrazole and isoniazid-induced epileptic seizure models	β-caryophyllene	Improved seizure activity	[189]
		Pentylenetetrazole (PTZ) or methyl-6,7-dimethoxy-4-ethyl-beta-carboline-3-carboxylate (DMCM)	HU308	No antiseizure effect	[190]
			AM630	Did not increase seizure activity	
		Pentylenetetrazole (PTZ)	AM1241	Increased seizure intensity	[191]
		Pentylenetetrazole (PTZ)	*Cnr2−/−*	Increased susceptibility to seizure	[192]
			JWH133	Did not alter seizure susceptibility	
Traumatic Brain Injury	Preclinical	Controlled cortical impact (CCI)	O-1966	Attenuated blood-brain barrier disruption and neural degeneration	[194]
				Induced acute neuroprotection	[195]
		Experimental closed-head injury (CHI)	HU-910	Enhanced neuroprotection and neurobehavioral recovery	[196]
		Experimental closed-head injury (CHI)		Increased expression of CB2Rs	[59]
		Controlled cortical impact (CCI)	JWH133	Reduced white matter injury	[197]

## 5. Conclusions and Future Therapeutic Perspectives

The debunking of the myth that CB2Rs are only found in the “periphery” and the absence of intoxicating effects of these receptors significantly increased the interest in the investigation of these receptors as a therapeutic target in neuropsychiatric and neurodegenerative disorders. Currently, we cannot completely be satisfied with the partially effective treatments for most of these disorders, and hence more research is warranted for new therapeutic targets using novel strategies, including artificial intelligence (AI), that might have better prediction and therapeutic outcomes with minimal adverse effects. The rapid explosion of cannabinoid research may also transform the current landscape by using in vitro and in vivo technologies that will enhance our understanding of the potential role of CBRs in neuropsychiatric and neurodegenerative disorders. Despite the recent progress, the withdrawal of the FAAH inhibitor during clinical trials and the adverse effects due to the anti-obesity CB1R antagonist have engendered cautionary approaches in clinical trials. It is encouraging that clinical trials targeting CB2R, and the approval by the Food and Drug Administration (FDA) of nabiximols for epileptic seizures, are turning the targeting components of the CB2R-ECS into a pharmacotherapeutic strategy. The findings included in this review showed that CB2Rs are highly expressed in neuropsychiatric and neurodegenerative disorders, and that selective CB2R ligands have promising effects on the symptomatic management of these disorders. Additional studies are required to evaluate the involvement of CB2Rs in these disorders using the full range of tools that are available to study the CB2Rs and their selective ligands in animal models as well as in controlled clinical trials. It is important that such future studies include translational and clinical profiles and in vivo and in vitro models expressing human CB2Rs. Furthermore, a careful evaluation of the side effects associated with the chronic treatment of CB2R ligands will provide further insights into the potential role of CB2Rs in regulating neurophysiological and behavioral functions.

The growing research interests in eCBome, the disturbances in its regulation, altered levels of eCBs, and discoveries of associated genetic polymorphisms provide targets for therapeutic intervention in neuropsychiatric and neurodegenerative disorders. Indeed, the densities of eCB receptors and the levels of eCBs have been suggested as possible biomarkers in neuropsychiatric disorders. With the increasing evidence that CB2Rs are involved in the brain, the functional role of the CB2R neuro-immune axis in pathophysiological signaling in the development of psychiatric disorders warrants further investigation. It is also noteworthy that the determination of the crystal structures of CB1R and CB2R reveals a yin-yang relationship and a functional profile of CB2R antagonism versus CB1R agonism [43,133,198]. This highlights another area where CB1R and CB2R seem likely to work both independently and/or cooperatively, and that will benefit from the development of critical medical applications. There are limitations and concerns regarding the use of CB2R medication for neurological disorders, as they are abundantly expressed in the periphery and may have peripheral side effects, while they may be useful in CNS disorders associated with neuroinflammation. It should be emphasized that a direct pharmacologic or genetic manipulation of the ECS might produce an effect different than the ones produced by phytocannabinoids. Further, research evaluating the numerous compounds in cannabis, along with terpenes and flavonoids, will add to our understanding of this natural eCBome in neuropsychiatric and neurodegenerative disorders and contribute to novel biomarkers and therapeutic agents.

## Figures and Tables

**Figure 1 ijms-23-00975-f001:**
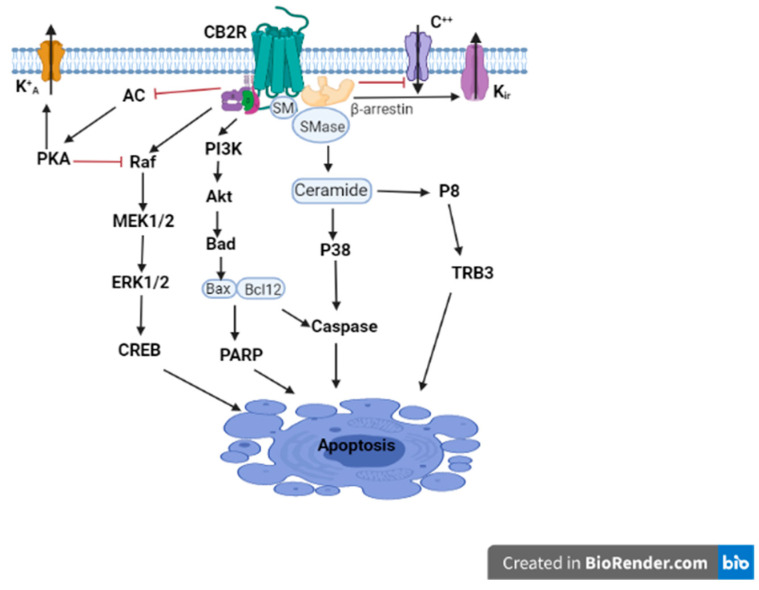
CB2R signaling. CB2R activation is associated with the Gβγ-dependent activation of the different MAPK cascades, and the Gαi/o-dependent inhibition of AC activity, in turn, results in the inhibition of A-type potassium channels (K + A). CB2R activation also results in the inhibition of specific calcium channels (Ca++), enhances the opening of inwardly rectifying potassium (Kir) channels, and stimulates the de novo synthesis of ceramide and the recruitment of β-arrestin. This figure is only a depiction of one outcome; these pathways can have multiple outcomes—ranging from stimulation of growth, manipulation of metabolism, control of the immune system to cell death—and the outcome is very context- and cell-dependent. The modulation of calcium alone can have multiple outcomes.

**Figure 2 ijms-23-00975-f002:**
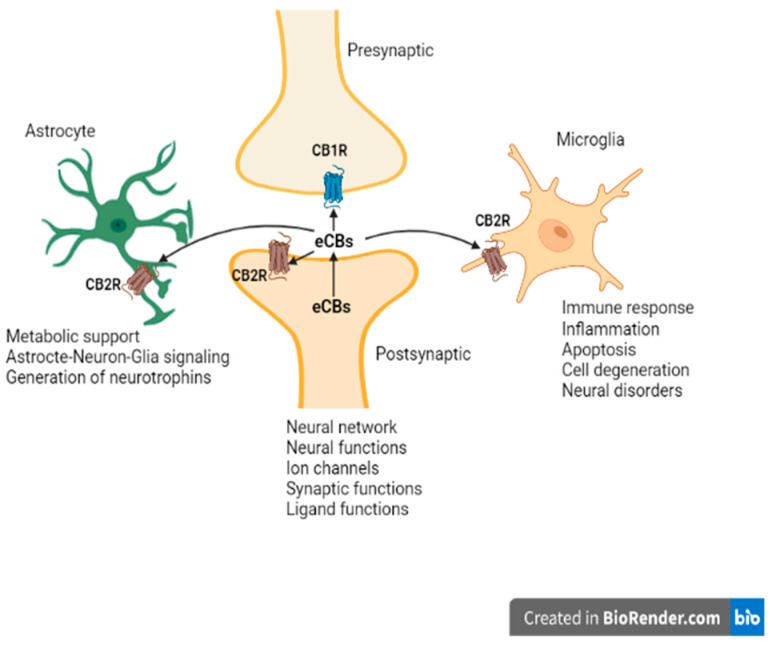
Endocannabinoid system CB2R’s neuro-immune crosstalk in neuropsychiatric disorders. We have used both IBA1 and CD11b antibodies, as IBA1 is a good marker that does not cross-react with neurons and astrocytes, while CD11b is a good marker for changes in microglia morphology.

## Data Availability

Not applicable.

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
