# Peer review of "New Insights and Potential Therapeutic Targeting of CB2 Cannabinoid Receptors in CNS Disorders"

_ijms, 2022, doi:10.3390/ijms23020975_

Round 1

Reviewer 1 Report

The paper is good written, it is easy for reading and the English is correct. The presentation is logical and systematic.

Author Response

Response to Reviewer 1 Comments

Comments: The paper is good written, it is easy for reading and the English is correct. The presentation is logical and systematic.

Response: We acknowledge reviewer 1 comments and time to review the manuscript.  We also appreciate reviewer 1 comments that the presentation is logical and systematic.

Reviewer 2 Report

The authors set out in this paper to review the function of CB2 receptors in the CNS and suggest that they could potentially be a pharmaceutical target. The underlying premise leading to the need to write the paper is that new evidence is suggesting these receptors are indeed found in the CNS, but before a certain point in time, the were thought not to be present, so providing the opportunity to write the paper. However, it is somewhat biased towards the concept of one drug on one receptor will fix complex pathophysiology, whereas the reality is that the pathophysiology itself is not understood, nor is the contextual role of the receptor itself in homeostasis. Plus, no drug is ever completely specific, so “off target” effects can also be just as important. Furthermore, most biological systems have multiple redundancy, so this has to be born in mind when discussing specific targeting. However, apart from these biases, the paper is a solid review of the subject, and is certainly reasonably comprehensive and mostly well written, apart from the abstract and introduction. In short, when the paper is just reviewing the literature, it is on stronger ground, but the interpretative bits are a bit biased. Perhaps a bit more balance?

Specific points:

  • Lines 12 & 17 – poor English
  • Line 34: need to make the point that most phytocannabinoids only interact weakly with the ECS (in effect, most of their mode of action lies in modulating systems outside the ECS)
  • Line 35: just how many psychoactive phytocannabinoids are there?!?!?
  • Line 41: should elaborate on the precise function of the ECS – as it stands it as an artificial construct defined by the fact that THC determined the first discovered target – it can be viewed as simply another component of a homeostatic mechanism to resist stress
  • Line 43: grammar (pluralisation)
  • Line 58: what do the authors mean by “unravelling” - very imprecise
  • Line 59: new paragraph?
  • Line 64: poor sentence construction
  • Line 69: might need to elaborate on the CNS immune system
  • Line 117: be very careful about making assumptions about the pathophysiology, in many cases, if not all, we still don’t fully understand it
  • Line 131: in section 2 may need to discuss a bit more about the central role of neuroinflammation in just about every CNS disease – one function of the CB2R is in modulating inflammation
  • Line 167: must emphasise the point that no drug is totally specific – must remember that “specificity” is something that is perhaps more important to the drug industry than how the underlying biology actually works
  • Line 169: Figure 1 is only really a depiction of one outcome; these pathways can have multiple outcomes, ranging from stimulation of growth, manipulation of metabolism, control of the immune system to cell death – and the outcome is very context and cell dependent. The modulation of calcium alone can have multiple outcomes
  • Line 283 onwards: need to elaborate on biphasic effects, every system in the body displays this, including both phytocannabinoid action and the actions of the ECS: links into what could be shown in figure 1. Low amounts of signal strength, for instance, through this pathway can have a very different outcome to high activation
  • Line 415: poor grammar
  • Line 452: in the preceding section need to be very careful about interpreting actions of the phytocannabinoids equalling actions on the ECS; this is not a truism at all. One can only say modifying the ECS when using drugs that preferentially bind components of the recognised ECS; however, even here, one has to remember that the ECS is an artificial construct built upon the first receptor identified that THC bound to. For example, THC powerfully modulates mitochondrial function.
  • Line 459: be careful, the ECS, like many other systems in the body, reacts to the causative agent of the metabolic syndrome, a poor lifestyle. In most cases it is not the cause; although there might be mutations within it that might accentuate it, for instance, enhancing appetite or reducing its ability to suppress inflammation.
  • Line 467: the causes of AN and BN are likely very different from the metabolic syndrome, so need to differentiate, although as in point 17, there could be a role of the ECS as it is a key system in energy and behaviour management.
  • Line 482: the key point here is that many modern conditions have an inflammatory basis, which, in most cases, are caused by a poor lifestyle. There is a very powerful link between inflammation, behaviour and many psychiatric conditions. The basis for this is pretty well understood.
  • Line 655: be very careful about making any direct claims between the CB2 receptor and epilepsy, except to say that progressive inflammation is a key pathology in this disease. How CBD works (and possibly THC, at the right dose), is not fully understood, but very likely involves interaction with multiple targes, ranging from TRPV1 to VDAC1 – and perhaps, modulation of oxidative stress.
  • Line 686: there are big lessons to be learnt from rimonabant, the most important of which was a complete misunderstanding of the role of the ECS beyond energy balance. Plus, no one likes to talk about the off-target effects of this drug.
  • Line 698: good, the key message is that we know a lot less than we think we do.
  • Line 715: key point – phytocannabinoid treatment does not equal direct manipulation of the ECS.

Author Response

Response to Reviewer 2 Comments

General comment: The authors set out in this paper to review the function of CB2 receptors in the CNS and suggest that they could potentially be a pharmaceutical target. The underlying premise leading to the need to write the paper is that new evidence is suggesting these receptors are indeed found in the CNS, but before a certain point in time, they were thought not to be present, so providing the opportunity to write the paper. However, it is somewhat biased towards the concept of one drug on one receptor will fix complex pathophysiology, whereas the reality is that the pathophysiology itself is not understood, nor is the contextual role of the receptor itself in homeostasis. Plus, no drug is ever completely specific, so “off target” effects can also be just as important. Furthermore, most biological systems have multiple redundancy, so this has to be born in mind when discussing specific targeting. However, apart from these biases, the paper is a solid review of the subject, and is certainly reasonably comprehensive and mostly well written, apart from the abstract and introduction. In short, when the paper is just reviewing the literature, it is on stronger ground, but the interpretative bits are a bit biased. Perhaps a bit more balance?

Response: First we acknowledge and thank reviewer 2's comments that has enhanced our review.  We agree with the comments and suggestions of the reviewer. There is no absolute selectivity and exogenously administered compounds may have off-target effects that ultimately result in unwanted adverse effects. However, compared to the CB1Rs, CB2Rs are associated with fewer adverse effects.   CB2R expression and their up-regulation has been associated with CNS disorders that are linked with underlying neuroinflammation.  The approval and use of epidiolex devoid of CNS intoxication, in epileptic seizures illustrates the possibility of targeting eCB-Gut-Immune-brain axis.  

Specific points:

Lines 12 & 17 – poor English

Response: Lines 12, 16, 17 and 18 have been re-worded.

Line 34: need to make the point that most phytocannabinoids only interact weakly with the ECS (in effect, most of their mode of action lies in modulating systems outside the ECS)

Response: Lines 34-36 have been re-worded

Line 35: just how many psychoactive phytocannabinoids are there?!?!?

Response: While Δ9-THC is the major psychoactive component as we highlighted, different authors’ report different numbers of phytocannabinoids including terpenes and flavonoids with Basavarajappa et al. (2017), reporting over 500 compounds present in the plant Cannabis sativa

Line 41: should elaborate on the precise function of the ECS – as it stands it as an artificial construct defined by the fact that THC determined the first discovered target – it can be viewed as simply another component of a homeostatic mechanism to resist stress

Response: The ECS is an endogenous signaling system involved in regulation and homeostasis in some physiological processes in the body, including neuro-immune communication between cells, appetite and metabolism, memory, and more. It is through this system of receptors and metabolic enzymes that cannabinoids interact with the human body and are intricately involved in reproduction and development. Lines 41-43.

Line 43: grammar (pluralisation)

Response: The grammar was reworded now in Line 47.

Line 58: what do the authors mean by “unravelling” - very imprecise

Response: The sentence was reconstructed precisely without the word “unravelling” now in  Lines 61-63.

Line 59: new paragraph?

Response: New paragraph inserted

Line 64: poor sentence construction

Response: The sentence was reconstructed now in Lines 68-69.

Line 69: might need to elaborate on the CNS immune system

Response: CB2R neuro-immune axis.  Many investigators were not able to detect the presence of neuronal CB2Rs in healthy brains (Munro et al., 1993; Galiegue et al., 1995; Griffin et al., 1999), but CB2R expression was demonstrated in rat microglia cells and other brain-associated immune cells during inflammation (Ibrahim et al., 2003; Benito et al., 2003, 2005; Golech et al., 2004; Nunez et al., 2004; Sheng et al., 2005). The expression of CB2Rs in brain immune cells like the microglia never ignited debate. Despite the evidence that CB2Rs might be present in the CNS, the expression of neuronal CB2Rs in the CNS has been much less well established and characterized in comparison with the expression of abundant brain CB1Rs.

Line 117: be very careful about making assumptions about the pathophysiology, in many cases, if not all, we still don’t fully understand it

Response: We endorse the premise that further preclinical and clinical research has to be done to fully understand the role of the ECS-CB2Rs in the pathophysiology of neuropsychiatric disorders. Lines 132-134.

Line 131: in section 2 may need to discuss a bit more about the central role of neuroinflammation in just about every CNS disease – one function of the CB2R is in modulating inflammation

Response: We have included a brief discussion on the role of neuroinflammation in CNS pathologies. Lines 130-138.

Line 167: must emphasise the point that no drug is totally specific – must remember that “specificity” is something that is perhaps more important to the drug industry than how the underlying biology actually works

Response: We agree with reviewers comments and throughout this review we have emphasized the inducible nature of CB2Rs during events with underlying inflammation.  This invariable makes CB2R a potential therapeutic target and ligands that activate or inhibit the activity of CB2Rs might be used to treat different disorders without causing adverse drug and intoxicating effects.

Line 169: Figure 1 is only really a depiction of one outcome; these pathways can have multiple outcomes, ranging from stimulation of growth, manipulation of metabolism, control of the immune system to cell death – and the outcome is very context and cell dependent. The modulation of calcium alone can have multiple outcomes

Response: The comment was addressed directly in figure 1 legend to include that the figure depicts one outcome and that the pathways can have multiple outcomes.

Line 283 onwards: need to elaborate on biphasic effects, every system in the body displays this, including both phytocannabinoid action and the actions of the ECS: links into what could be shown in figure 1. Low amounts of signal strength, for instance, through this pathway can have a very different outcome to high activation

Response: The nature of biphasic effects of phytocannabinoids in various biological system are common responses observed after administration cannabinoids in pre-clinical, clinical and anecdotal reports. Throughout the manuscript we tried to explain such responses based on evidence from the literature.

Line 415: poor grammar

Response: The sentence was revised as suggested. Lines 419-420.

Line 452: in the preceding section need to be very careful about interpreting actions of the phytocannabinoids equalling actions on the ECS; this is not a truism at all. One can only say modifying the ECS when using drugs that preferentially bind components of the recognised ECS; however, even here, one has to remember that the ECS is an artificial construct built upon the first receptor identified that THC bound to. For example, THC powerfully modulates mitochondrial function.

Response: This is an important point and throughout the manuscript we emphasize that modulation of the ECS specially the CB2Rs might serve as a potential therapeutic target for the treatment of CNS disorders especially in conditions with underlying neuroinflammation.  The emerging eCB-CB2R-Gut-immune-brain axis requires further investigations (basic and/or clinical) in understanding the underlining mechanisms of CB2Rs as immunomodulatory and neuromodulatory receptor in CNS pathologies for which there is upregulation of CB2Rs.

Line 459: be careful, the ECS, like many other systems in the body, reacts to the causative agent of the metabolic syndrome, a poor lifestyle. In most cases it is not the cause; although there might be mutations within it that might accentuate it, for instance, enhancing appetite or reducing its ability to suppress inflammation.

Response: We agree on the fact that metabolic disorders like diabetes are mainly linked with life style and the main strategy in the management of such disorders is life style modifications. Here we tried to highlight the existence of a link between the ECS and metabolic disorders and targeting the ECS might help fight metabolic disorders. We included the comments of the referee in the revised manuscript. Lines 464-466.

Line 467: the causes of AN and BN are likely very different from the metabolic syndrome, so need to differentiate, although as in point 17, there could be a role of the ECS as it is a key system in energy and behaviour management.

Response: We agree that the causes of AN and BN are different from metabolic disorders. It would have been interesting to explore the detailed causes of AN and BN and their association with metabolic disorders. However, in the case of this review we provide a general concept about AN and BN and, as the referee pointed out, we tried to show the link between the ECS and energy metabolism. Despite contrasting results, researchers have examined the association of the three major eating disorders—anorexia nervosa, bulimia nervosa, and binge-eating disorder with metabolic syndrome.

Line 482: the key point here is that many modern conditions have an inflammatory basis, which, in most cases, are caused by a poor lifestyle. There is a very powerful link between inflammation, behaviour and many psychiatric conditions. The basis for this is pretty well understood.

Response: We agree with the reviewer’s assessment and noted in the review.

Line 655: be very careful about making any direct claims between the CB2 receptor and epilepsy, except to say that progressive inflammation is a key pathology in this disease. How CBD works (and possibly THC, at the right dose), is not fully understood, but very likely involves interaction with multiple targes, ranging from TRPV1 to VDAC1 – and perhaps, modulation of oxidative stress.

Response: We have incorporated this view in the review.

Line 686: there are big lessons to be learnt from rimonabant, the most important of which was a complete misunderstanding of the role of the ECS beyond energy balance. Plus, no one likes to talk about the off-target effects of this drug.

Response: We have discussed the issue of rimonabant elsewhere, and the failed clinical trial of FAAH inhibitor in this review in Line 697.

Line 698: good, the key message is that we know a lot less than we think we do.

Response: We agree and more studies are required as we emphasized throughout the review.

Line 715: key point – phytocannabinoid treatment does not equal direct manipulation of the ECS.

Response: This is an excellent suggestion and was included in the revised manuscript. Lines 723-725.

Reviewer 3 Report

This work reviews recent advances in the role of CB2R in neuropsychiatric and neurodegenerative disorders, including but not limited to anxiety, depression, schizophrenia, Parkinson's disease (PD), Alzheimer's disease (AD), Huntington's disease (HD) and addiction. The cannabinoid receptor (CB2R) subtype 2 represents an interesting and new therapeutic target for its involvement in the first stages of neurodegeneration as well as in the onset and progression of cancer.

The authors paid a lot of attention to the mechanisms of action of CB2R and in this review showed that CB2Rs are highly expressed in neuropsychiatric and neurodegenerative disorders, and that selective CB2R ligands have promising effect in the symptomatic treatment of these disorders.

According to the authors, there are limitations and concerns to the use of CB2R drugs in neurological disorders as they are expressed abundantly in the periphery and may have peripheral side effects, while they may be useful in neuritis-related CNS disorders. Moreover, studies evaluating the numerous compounds found in cannabis along with terpenes and flavonoids will deepen our understanding of this natural eCBome in neuropsychiatric and neurodegenerative disorders and contribute to the development of new biomarkers and therapeutic agents in health.

The manuscript submitted for review has a high cognitive value and broadens our understanding of the biological mechanisms of CB2R. The cited results and their interpretation do not raise any major concerns. Therefore, I propose to publish the article as it is.

Author Response

The manuscript submitted for review has a high cognitive value and broadens our understanding of the biological mechanisms of CB2R. The cited results and their interpretation do not raise any major concerns. Therefore, I propose to publish the article as it is.

Response: We appreciate the time and effort that this reviewer dedicated to providing valuable feedback on our manuscript. As this reviewer clearly mentioned CB2Rs are currently becoming new therapeutic targets in the search for new drugs for the treatment of psychiatric disorders. Unlike the CB1Rs which are associated with different adverse effects in the CNS, CB2Rs are devoid of such adverse effects and might serve as a new therapeutic targets.  This requires further pre-clinical studies and clinical trials.